# Oligonucleotide Therapies for Facioscapulohumeral Muscular Dystrophy: Current Preclinical Landscape

**DOI:** 10.3390/ijms25169065

**Published:** 2024-08-21

**Authors:** Samuel L. Beck, Toshifumi Yokota

**Affiliations:** 1Department of Biological Sciences, Faculty of Science, University of Alberta, Edmonton, AB T6G 2R3, Canada; sbeck1@ualberta.ca; 2Department of Medical Genetics, Faculty of Medicine and Dentistry, University of Alberta, Edmonton, AB T6G 2R3, Canada

**Keywords:** facioscapulohumeral muscular dystrophy (FSHD), DUX4, antisense oligonucleotides, RNAi, CRISPR, DNA aptamers, DNA decoys, targeted therapy, genetic therapy

## Abstract

Facioscapulohumeral muscular dystrophy (FSHD) is an inherited myopathy, characterized by progressive and asymmetric muscle atrophy, primarily affecting muscles of the face, shoulder girdle, and upper arms before affecting muscles of the lower extremities with age and greater disease severity. FSHD is a disabling condition, and patients may also present with various extramuscular symptoms. FSHD is caused by the aberrant expression of double homeobox 4 (*DUX4*) in skeletal muscle, arising from compromised epigenetic repression of the D4Z4 array. *DUX4* encodes the DUX4 protein, a transcription factor that activates myotoxic gene programs to produce the FSHD pathology. Therefore, sequence-specific oligonucleotides aimed at reducing DUX4 levels in patients is a compelling therapeutic approach, and one that has received considerable research interest over the last decade. This review aims to describe the current preclinical landscape of oligonucleotide therapies for FSHD. This includes outlining the mechanism of action of each therapy and summarizing the preclinical results obtained regarding their efficacy in cellular and/or murine disease models. The scope of this review is limited to oligonucleotide-based therapies that inhibit the *DUX4* gene, mRNA, or protein in a way that does not involve gene editing.

## 1. Introduction

### 1.1. FSHD Overview

Facioscapulohumeral muscular dystrophy (FSHD, MIM: 158900 and 158901) is an autosomal dominant myopathy with a global incidence of approximately 1 in 8000–22,000, making it one of the most common forms of muscular dystrophy worldwide [1,2,3]. FSHD is characterized by progressive muscle weakness and atrophy that develops in a left–right asymmetric fashion, primarily affecting muscles of the face, shoulder girdle, and upper arms. Additional muscle groups can be affected with age, such as the ankle dorsiflexors and proximal leg muscles, resulting in obligate wheelchair use for approximately 20% of patients [1,4]. Some FSHD patients also experience extramuscular symptoms such as hearing loss, retinal vasculopathy, and/or cardiac conduction defects. FSHD is highly variable in terms of disease onset and severity [4]. There are no curative treatments available for FSHD, with current interventions limited to managing symptoms [5].

FSHD is a genetic condition with two distinct types. Patients can be classified as having either FSHD1 or FSHD2 depending on the genetic mechanism that results in the de-repression of the D4Z4 macrosatellite repeat array, located in the subtelomeric region 4q35 (Figure 1A). Healthy individuals have 11–100 + 3.3 kb D4Z4 repeat units. FSHD1, affecting 95% of patients, is caused by array contraction to ≤10 repeats [6]. FSHD2, affecting 5% of patients, is caused by a mutation in genes involved in epigenetic methylation of the D4Z4 array (e.g., *SMCHD1, DNMT3B*, and *LRIF1*) [7,8,9]. FSHD1 is diagnosed by assessing the size of the repeat contraction, while FSHD2 diagnosis also requires testing for a mutation in *SMCHD1, DNMT3B,* and/or *LRIF1* [4]. Curiously, FSHD2 patients also tend to have fewer D4Z4 repeats than healthy individuals (12–16), demonstrating the complexity of this condition and how the distinction between FSHD1 and FSHD2 may not be as straightforward as initially thought [10]. Regardless, the shared outcome in both types of FSHD is the loss of repressive methylation in the D4Z4 array. Therefore, the clinical presentation is identical between FSHD1 and FSHD2 [4].

### 1.2. DUX4 Is the Central Cause of FSHD

FSHD is caused by the aberrant expression of the double homeobox 4 (DUX4) protein in skeletal muscle, arising from the loss of chromatin repression at the D4Z4 array. The *DUX4* gene encodes a transcription factor that is normally involved in zygotic genome activation during the four-cell stage of early embryonic development [11,12]. Afterward, *DUX4* is epigenetically silenced in all adult tissues apart from the limited expression of unrelated *DUX4* isoforms in the testis and thymus [13,14]. Alternative splicing is known to occur for the *DUX4* transcript; however, only mis-expression of the full-length isoform in muscle tissue is relevant to FSHD. Any mention hereafter of *DUX4* mRNA refers only to this full-length, pathogenic isoform.

Narrowing in on the genetic region from which FSHD arises, we return to the D4Z4 repeat array and the *DUX4* gene therein. Within each 3.3 kb D4Z4 repeat unit is a retrogene containing exons 1 and 2 of the *DUX4* open reading frame. A partial D4Z4 unit occurs after the most distal complete unit, followed by the 3rd *DUX4* exon (Figure 1B) [15]. Aberrant *DUX4* expression occurs from this distal D4Z4 unit in the FSHD-permissive 4qA haplotype. FSHD can only manifest in one of two major 4q allele variants: 4qA and 4qB. Unlike the non-permissive 4qB haplotype, the 4qA haplotype contains the pLAM region with a polyadenylation site (PAS), allowing for the transcription of a stable *DUX4* mRNA when epigenetic repression is compromised in the D4Z4 array [1,15].

Following transcription of the *DUX4* gene, stochastic, low-level DUX4 protein expression occurs in the myofiber nuclei of the skeletal muscle. Inappropriate DUX4 protein expression in adult skeletal muscle is highly toxic, driving gene programs that result in oxidative stress, dysregulated transcript quality control, protein aggregation, inflammation, apoptosis, impaired myogenesis, and muscle atrophy (Figure 1B) [15,16,17]. The DUX4 protein directly activates various genes including *TRIM43*, *ZSCAN4*, *MBD3L2*, *WFDC3*, *PRAMEF1*, *RFPL2*, and *KHDC1* [18,19]. Disruption of these signaling pathways by reactivated DUX4 produces the FSHD pathology that we see in patients, manifesting primarily in the muscle tissue. This makes mis-expressed DUX4 the primary therapeutic target for treating FSHD. Sequence-specific oligonucleotides comprise a large portion of targeted therapies that are currently under investigation for FSHD.

### 1.3. DUX4-Related Challenges for Preclinical Research

While DUX4 is a convenient therapeutic target for FSHD, certain DUX4 characteristics can make it challenging to evaluate treatment efficacy in preclinical experiments. One notable challenge is that the DUX4 protein is very difficult to detect in patient muscle tissue due to its low expression level and infrequent DUX4-positive myonuclei [13]. As a result, outcome measures concerning treatment efficacy tend to vary between studies. While *DUX4* mRNA levels are usually measured to determine post-treatment knockdown, studies often use indirect measures as well, such as DUX4 target gene expression [19].

Another important consideration is that the D4Z4 macrosatellite region and the *DUX4* gene are restricted to Old World primates, making animal models, murine and otherwise, incapable of exactly recapitulating the expression characteristics and disease phenotype seen in human FSHD patients [20]. To address this, *DUX4* expression must be artificially introduced in these mammalian models. Similarly, non-patient cell models must also introduce *DUX4* to evaluate its knockdown by targeted oligonucleotides. However, patient-derived primary or immortalized cell lines, being the most common cellular FSHD model, maintain low-level *DUX4* expression [21].

### 1.4. Delivery Methods for DUX4-Targeting Oligonucleotides

As of the writing of this review, several delivery strategies have been used in preclinical studies with *DUX4*-targeting oligonucleotides. These include Vivo conjugation, fatty acid conjugation (palmitoyl), and adeno-associated virus (AAV) vectors. Firstly, Vivo octa-guanidine dendrimers are a synthetic, cell-penetrating molecule used for the delivery of antisense phosphorodiamidate morpholino oligomers (PMOs) [22]. Palmitoyl, a type of fatty acid conjugation, is used on antisense oligonucleotides to improve muscle delivery and potency [23]. Lastly, AAV vectors are small, non-pathogenic DNA viruses capable of efficiently transducing cells for sustained expression of unmodified oligos, such as shRNAs or miRNAs [24].

## 2. Oligonucleotide Therapies Targeting DUX4

FSHD is a condition arising solely from the aberrant reactivation of a dormant gene: *DUX4*. Therefore, therapies that directly target aberrant *DUX4* expression present a compelling treatment option. This review focuses on preclinical FSHD therapies that use a sequence-specific approach for targeting DUX4. This includes oligonucleotides with sequence complementarity to either the *DUX4* gene, *DUX4* mRNA, or the DUX4 protein. This complementarity is used to inhibit *DUX4* somewhere along its gene > mRNA > protein expression axis, thereby preventing DUX4 transactivation and the resulting FSHD pathology (Figure 2). In this review, we discuss the different types of targeted oligonucleotide therapies for FSHD, summarize the preclinical results obtained so far, and discuss further considerations for these treatment approaches. Notably, the scope of this review is limited to only non-gene-editing approaches that target DUX4.

The targeted oligonucleotide therapies discussed in this review are divided into three categories: antisense oligonucleotides (AOs), RNA interference (RNAi), and other oligonucleotides. These therapies have shown promising preclinical results in cellular and murine models of FSHD, primarily in their ability to lower *DUX4* mRNA levels, reduce DUX4 target gene expression, and alleviate FSHD symptoms (Table 1, Table 2 and Table 3).

### 2.1. Antisense Oligonucleotides (AOs)

Antisense oligonucleotides (AOs) are synthetic, single-stranded nucleic acids that target a complementary mRNA molecule, dictated by Watson–Crick base pairing, to initiate post-transcriptional gene silencing. AOs were first identified in 1978 by Zamecnik and Stephenson, who found that complementary oligonucleotides inhibited the translation of Rous sarcoma virus mRNA [59]. AOs bind to a target mRNA in a sequence-dependent manner and prevent its translation, thereby reducing the amount of the target protein [60].

AOs are known to utilize various chemistries, a fact that makes them distinct from other oligonucleotide therapies. Modern AO drugs have chemical modifications to improve their pharmacological properties like tolerability, target affinity, nuclease resistance, and intracellular uptake [61,62]. Commonly used AO chemistries involve modifying the phosphate backbone (PS, phosphorothioate; PMO, phosphorodiamidate morpholino oligomer) or ribose sugar (2′OMe, 2′-*O*-methyl; 2′-MOE, 2′-*O*-methoxyethyl; LNA, locked nucleic acid) [61]. In addition, dendrimer and fatty acid conjugate modifications have been used to facilitate the delivery of *DUX4*-targeting AOs [29,33]. AOs can also be synthesized as gapmers, a chimeric molecule comprised of a central DNA region and a flanking region of modified RNA [63].

The formation of an AO-mRNA duplex results in (1) RNase H-mediated degradation of the target mRNA or (2) steric blocking of the target mRNA (Figure 3A) [60,62,64]. Only gapmer AOs recruit RNase H to cleave the target mRNA. Gapmer AOs produce a DNA–RNA substrate that when bound to mRNA is recognizable by RNase H [65,66]. Steric-blocking AOs inhibit proper mRNA translation, splicing, and/or stability in an RNase H-independent manner [60,62,64]. Additionally, these steric-blocking AOs can initiate further downstream degradation pathways, such as nonsense-mediated decay and no-go decay [67,68].

As summarised in Table 1, numerous preclinical studies have demonstrated that AO therapies can effectively reduce the amount of *DUX4* mRNA and DUX4 target gene expression both in vitro and in vivo [25,26,27,28,29,30,31,32,33,34,35,36,37,38]. In particular, demonstrating *DUX4* knockdown in vivo represents a notable milestone in the preclinical development of AO therapies, something that has been recently achieved by several research groups [28,29,31,32,33,35,36,37,38]. Other measures were also used to evaluate AO treatment efficacy, such as DUX4 protein levels, muscle fiber health, and murine functional performance. Most groups used primary or immortalized myoblasts/myocytes (often differentiated into myotubes) as an in vitro FSHD model, and *FLExDUX4* mice as an in vivo FSHD model [69,70,71]. Earlier studies used local (intramuscular) injection for in vivo AO treatment, while studies after 2021 evaluated systemic (intraperitoneal, subcutaneous) injection routes. Systemic injection, being more clinically viable, managed to yield similarly efficient *DUX4* knockdown compared with local injection. Regarding the *DUX4* target site, the >10 studies produced since 2011 tend to target exons 2 and 3, often with an emphasis on the polyadenylations site (PAS), pre-mRNA cleavage sites, and/or splice sites therein (Figure 4). Exons 2 and 3 are preferred *DUX4* target sites because this region differs from the largely homologous *DUX4c* gene. This DUX4–DUX4c sequence homology concerns the N-terminal region at the DNA-binding homeobox domains, encoded by exon 1 [72].

### 2.2. RNA Interference (RNAi)

Like AOs, RNAi-based oligonucleotides act at the RNA level, binding to a target mRNA according to antisense sequence complementarity to initiate post-transcriptional gene silencing. Where these classifications differ is that RNAi-based oligonucleotides initiate the RNA interference (RNAi) pathway to knockdown target mRNAs. First defined by Fire et al. in 1998, RNAi is a conserved, biological mechanism by which double-stranded RNA triggers the loss of homologous mRNA [73]. RNAi can be induced by miRNAs or siRNAs complementary to an mRNA transcript. DICER endonucleases cleave precursor molecules (pre-miRNA or shRNA) to produce mature microRNA (miRNA) or small-interfering RNA (siRNA), which then are loaded into the Argonaute (AGO) protein of the RNA-induced silencing complex (RISC). Using the guide strand, the RISC targets a complementary mRNA transcript and induces translational inhibition, sequestration, and/or mRNA degradation (miRNA-RISC), or simply mRNA degradation (siRNA-RISC) (Figure 3B) [73,74].

The two types of RNAi-based oligonucleotide therapies for FSHD are miRNAs (natural or artificial) and siRNAs. Since 2011, several studies have shown that these RNAi-based oligos can knockdown *DUX4* mRNA and reduce DUX4 transactivation, in addition to improving other markers of FSHD symptom reversal (e.g., DUX4 protein levels, muscle fiber health, murine functional performance, etc.) (Table 2) [25,26,39,40,41,42,43,44,45,46,47,48]. Notably, the amount of published work is less for RNAi-based approaches compared with AOs. These studies commonly used primary or immortalized myoblasts/myocytes (often differentiated into myotubes) as an in vitro FSHD model, and AAV-DUX4 mice as an in vivo FSHD model [69,70,75]. *FLExDUX4* mice were also used as an in vivo model for testing RNAi therapies, but to a lesser extent [71]. Most studies used local (intramuscular) injection to evaluate preclinical efficacy in vivo, except for three groups with partially released findings using systemic (intravenous) injection [42,45,46,47,48]. All current preclinical studies opted for adeno-associated virus (AAV)-mediated delivery of siRNAs or miRNAs in vivo, except for the partially released findings from Avidity Biosciences, Inc. and Dyne Therapeutics, Inc., which each describe a proprietary anti-mTfR1 mAb conjugate for delivery [45,46,47,48]. Lastly, as summarized in Figure 4, these miRNAs and siRNAs target *DUX4* mRNA at all three of its exons, especially exon 1. Other sites like upstream D4Z4 regions, intronic regions, and pre-mRNA cleavage sites have also been tested.

### 2.3. Other Oligonucleotides

Other non-gene-editing, preclinical oligonucleotide therapies have also been investigated as potential treatments for FSHD (Table 3). Unlike AO and RNAi therapies which target *DUX4* mRNA, these oligos tend to target *DUX4* expression at the gene or protein level (Figure 2). These approaches present further compelling options for treating FSHD, in addition to the antisense approaches previously discussed.

#### 2.3.1. CRISPR/dCas9 Transcriptional Repression

Multiple research groups have explored CRISPR/dCas9-mediated transcriptional repression of the *DUX4* gene as a targeted therapy for FSHD. This is a form of CRISPR inhibition (CRISPRi) which uses the sequence specificity of the sgRNA-Cas9 complex to target the DUX4 promoter, but with a catalytically inactive ‘dead’ Cas9 (dCas9) fused to a transcriptional repressor domain (TRD) [76]. This allows for specific re-silencing of the D4Z4 region, reducing *DUX4* expression and the resulting FSHD pathology. While various TRDs have been used, most studies opted for the Krüppel-associated box (KRAB) domain. Evaluation of this treatment approach has been largely conducted in primary FSHD myoblasts, myocytes, or myotubes, with the only in vivo testing performed by Himeda et al. (2021) using AAV delivery and local (intramuscular) injection in *FLExDUX4* mice [53]. These studies have all reported a reduction in *DUX4* mRNA following treatment, as well as a reduction in DUX4 target gene expression and/or increased H3K9 tri-methylation at the D4Z4 array [49,50,53,54,58]. Notably, rather than directly repressing the *DUX4* gene, Himeda et al. (2018) used the CRISPR/dCas9-KRAB system to repress epigenetic activators of *DUX4* (*ASH1L*, *BRD2*, *KDM4C*, *SMARCA5*). A similar reduction in *DUX4* mRNA was observed for this approach [50].

Abstracts proposing other non-gene-editing, CRISPR-based approaches have been recently published. The results are preliminary and have not been fully released, however. One group is developing CRISPR/Cas13-mediated cleavage of *DUX4* mRNA, reporting effective *DUX4* knockdown in vivo [56]. Another group suggests using CRISPR/Cas13-ADAR (adenosine deaminase acting on RNA)-mediated editing of *DUX4* mRNA to create a C > U nonsense mutation [57]. No definitive results have been published at this time.

#### 2.3.2. DNA Aptamers

Aptamers are single-stranded oligonucleotides that can bind to a specific protein or protein family thanks to their secondary and tertiary folding structure. The unique 3D conformation of an aptamer allows for target interaction like that of an antigen and antibody [77]. Therefore, aptamers can be used to specifically target a protein of interest and, in the case of FSHD, bind to and inhibit DUX4. Klingler et al. (2020) designed DNA aptamers with a high affinity to the DUX4 protein [51]. While not evaluated, these DNA aptamers could be used to treat FSHD by sterically inhibiting the DUX4 protein in skeletal muscle.

#### 2.3.3. dsDNA Decoy Trapping

Mariot et al. (2020) demonstrated a unique approach to prevent DUX4 transactivation known as decoy trapping [52]. Decoy trapping uses double-stranded DNA fragments whose sequence corresponds to DUX4 binding motifs, akin to the DNA regions that DUX4 normally binds to as a transcription factor. By saturating the cellular environment with dsDNA decoy binding sites, the DUX4 protein is trapped in a binding sink and unable to activate its normal target genes. Mariot et al. (2020) found that dsDNA treatment was able to reduce the expression of downstream DUX4 target genes in vitro and in vivo [52].

#### 2.3.4. U7-snRNA Pre-mRNA Inhibition

Rashnonejad et al. (2021) describe a strategy to inhibit *DUX4* mRNA expression using U7-small nuclear RNA (snRNA) antisense expression cassettes [55]. U7-snRNA is a part of the small nuclear ribonucleoprotein complex (snRNP), which is involved in 3′ end processing of histone pre-mRNAs in the nucleus. This therapeutic approach uses modified U7-snRNA with antisense sequence specificity to *DUX4*, capable of inhibiting pre-mRNA production or maturation. Rashnonejad et al. (2021) showed that these U7-snRNA expression cassettes, delivered by AAV, effectively reduced *DUX4* mRNA, the DUX4 protein, and DUX4 target gene expression in immortalized FSHD myotubes [55].

## 3. Further Considerations

Antisense therapies for FSHD, meaning AOs or RNAi drugs that target *DUX4* mRNA, are the furthest along in preclinical development compared to other oligonucleotide approaches. Many AO and RNAi therapies have shown promising indications both in vitro and in vivo, as discussed previously. Therefore, this discussion of certain advantages and disadvantages of *DUX4*-targeting oligonucleotide therapies will focus on antisense approaches only.

### 3.1. Advantages of Antisense Approaches

Antisense therapies are ideal for monogenic diseases that can be attributed to a single root cause. In the case of FSHD, this is aberrant *DUX4* expression in skeletal muscle. Another advantage of antisense therapies is that they are highly specific and employ potent molecules with a relatively simple mechanism of action, often taking advantage of conserved cellular processes [60].

Additionally, compared with gene editing approaches that prevent *DUX4* expression, antisense therapies involve no changes to genomic DNA, acting only at the RNA level [60,64]. This makes them more acceptable from a regulatory standpoint, unburdened by the moral concern surrounding CRISPR/Cas9 editing of the human genome, even if for therapeutic purposes. For this reason, it may be fair to suggest that antisense therapies are a more clinically viable form of targeted, genetic therapy for FSHD.

### 3.2. Disadvantages of Antisense Approaches

Efficient delivery to muscle tissue is a considerable challenge for antisense therapies, often hindering the clinical utility of oligonucleotides that otherwise demonstrate good preclinical efficacy. AOs and RNAi oligonucleotides are relatively large nucleic acids that tend to be negatively charged and hydrophilic [61]. Molecules with such properties do not readily pass through the plasma membrane. Furthermore, upon systemic injection, these molecules must avoid nuclease degradation, mononuclear phagocyte system entrapment, protein entrapment, and high renal clearance [78,79,80]. If these can be overcome, there remains the issue of inefficient cellular uptake, as these oligonucleotides are also prone to endosomal entrapment within the cell [78,79,80]. All this means that only a small percentage of the injected drug becomes bioavailable to provide therapeutic benefit to a patient. However, various strategies to improve delivery are currently being explored for AOs and RNAi oligos, including chemical modification, delivery conjugates, and carrier molecules [78].

The persistence of the therapeutic effect is another notable disadvantage of antisense therapies. Given that antisense therapies act on mRNA, a transient and replenishable molecule, regular lifelong administrations may be necessary to offer long-term reversal of FSHD symptoms for patients. This problem is not shared by other proposed genetic therapies that would permanently inactivate the toxic *DUX4* gene (e.g., CRISPR/Cas9 editing) such that multiple treatments are not needed.

### 3.3. Early-Stage Clinical Trials for Select FSHD Therapies

Two RNAi-based oligonucleotide therapies for FSHD are currently recruiting for Phase 1/2 clinical trials to evaluate their safety, tolerability, pharmacokinetics, pharmacodynamics, and efficacy in adult patients. First, ARO-DUX4, developed by Arrowhead Pharmaceuticals, is a *DUX4*-specific siRNA using an unspecified and proprietary delivery method (Phase 1/2 NCT06131983) [81,82]. Second, AOC 1020, developed by Avidity Biosciences, is a *DUX4*-specific siRNA using a proprietary anti-mTfR1 mAb delivery conjugate (Phase 1/2 FORTITUDE™ NCT0574792) [83,84]. Both therapies have previously demonstrated preclinical efficacy in cellular and murine models of FSHD [42,43,45,46].

## 4. Conclusions

Since *DUX4* was identified as the central cause of FSHD, numerous targeted oligonucleotide therapies have been proposed, many of which have shown promising results in preclinical stages. However, despite *DUX4* presenting itself as an ideal therapeutic target, there are still considerable challenges that may prevent these therapies from reaching clinical use and benefiting patients. First, more progress towards fully characterizing FSHD is needed, as it remains an incredibly complicated condition with many unanswered questions. Further research into the molecular underpinnings of FSHD may offer additional therapeutic targets amenable to oligonucleotide therapies. Similarly, it is important to continue investigating the normal physiological role of DUX4 in the testis and thymus, as this is not fully understood and could impact decisions made when targeting *DUX4* in skeletal muscle, possibly in terms of off-target effects.

Another consideration would be the potential synergistic effect of combining multiple therapies, particularly those that target *DUX4* expression via different modes of action. While this has not yet been attempted for FSHD, combined therapies have been explored considerably for Duchenne muscular dystrophy (DMD), with certain combinations improving treatment efficiency. In particular, proposed combination treatment strategies for DMD often use one therapy that corrects the genetic defect (e.g., AO) and another that addresses secondary disease manifestations [85].

Overall, with a strong pipeline of candidate oligos from many different research groups and two siRNA drugs entering early clinical trials, the future appears hopeful for a targeted treatment option for patients with FSHD.

## Figures and Tables

**Figure 1 ijms-25-09065-f001:**
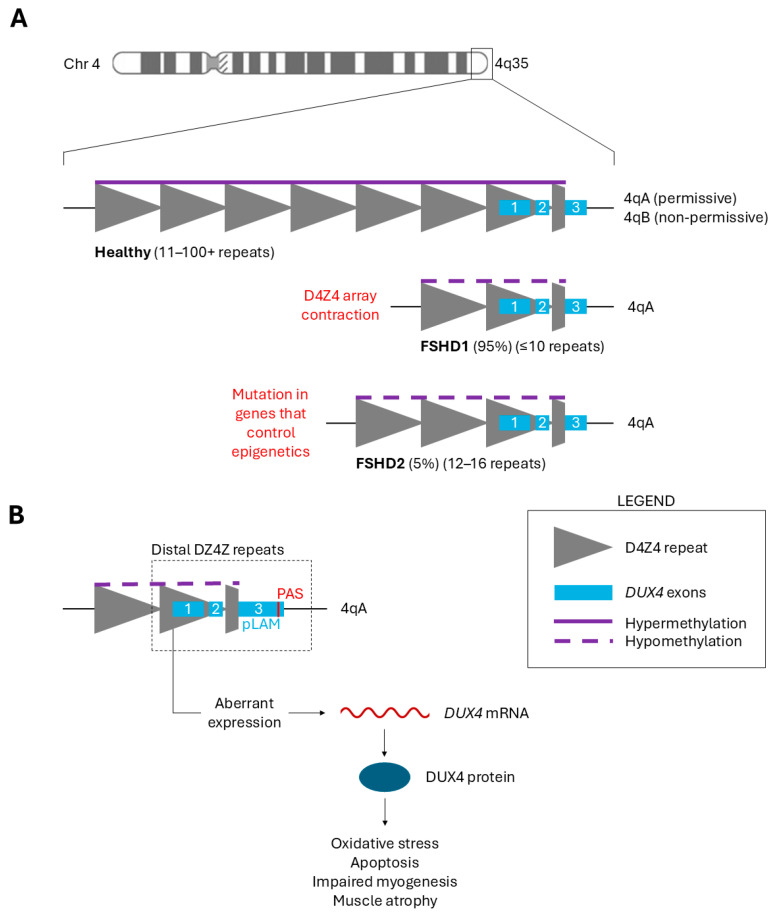
(**A**) A schematic representation of the D4Z4 region in healthy individuals and FSHD patients. The D4Z4 macrosatellite tandem repeat array is found in the subtelomeric region 4q35. Each gray triangle indicates a 3.3 kb D4Z4 repeat unit, within each of which a *DUX4* retrogene is contained. The 1st and 2nd *DUX4* exons (blue boxes) occur in each full D4Z4 unit. A partial D4Z4 (gray trapezoid) occurs after the most distal complete unit, followed by the 3rd *DUX4* exon. Healthy individuals have 11–100+ repeats and full epigenetic repression (purple line). FSHD patients have fewer repeats and compromised epigenetic repression (purple dotted line) arising from one of two genetic changes indicated in red text. (**B**) A schematic representation of aberrant *DUX4* expression from the most distal complete D4Z4 repeat within the FSHD-permissive 4qA haplotype. The 1st and 2nd *DUX4* exons are within each D4Z4 unit. The 3rd *DUX4* exon and PAS site are found directly downstream of the most distal D4Z4 unit. Compromised repression results in low-level DUX4 protein expression existing within the skeletal muscle, perturbing downstream gene expression to cause the FSHD pathology (oxidative stress, apoptosis, impaired myogenesis, muscle atrophy, etc.).

**Figure 2 ijms-25-09065-f002:**
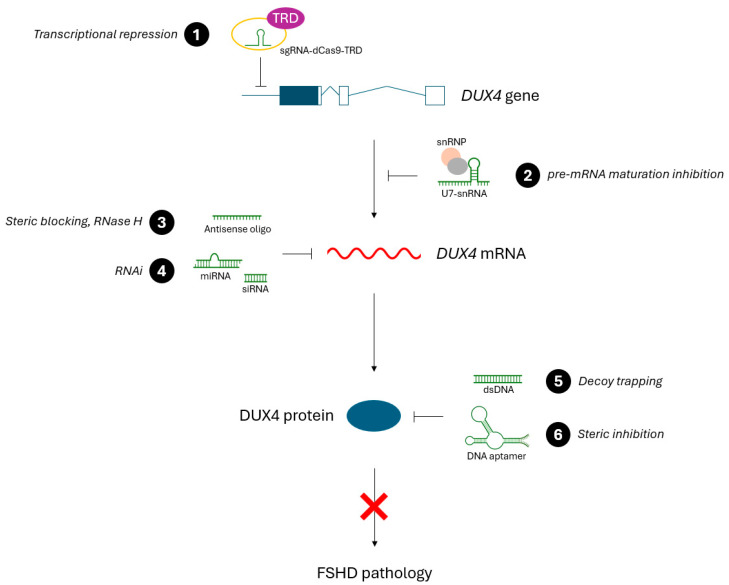
An overview of oligonucleotide therapies for FSHD and where they inhibit *DUX4* expression (gene, mRNA, or protein) to ameliorate FSHD symptoms. (**1**) sgRNA-dCas9-TRD targets the promoter or coding region of the *DUX4* gene, resulting in transcriptional repression. (**2**) U7-snRNA alters the specificity of a small nuclear ribonucleoprotein complex (snRNP) to inhibit *DUX4* pre-mRNA maturation. (**3**) Antisense oligonucleotides bind to *DUX4* mRNA, causing steric blocking (and downstream effects) or RNase H-mediated degradation, depending on the AO type. (**4**) Various RNA molecules (miRNA, siRNA, etc.) degrade *DUX4* mRNA through the RNA interference pathway. (**5**) Decoy dsDNA molecules have DUX4-binding motifs that trap the DUX4 protein in a binding sink, inhibiting the transactivation of downstream DUX4 targets. (**6**) DNA aptamers bind to the DUX4 protein, inhibiting DUX4 activity through steric inhibition.

**Figure 3 ijms-25-09065-f003:**
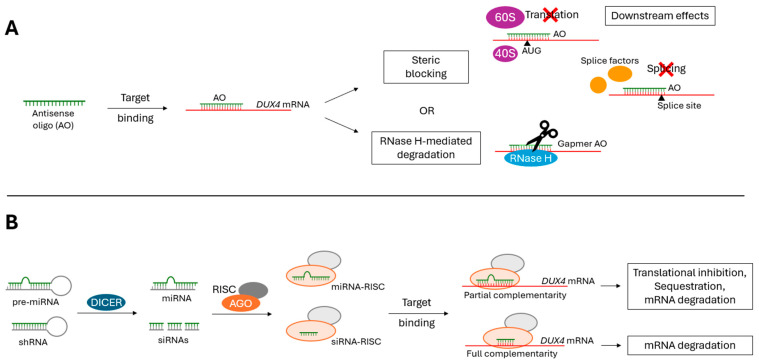
The mechanism of action for (**A**) AO and (**B**) RNAi therapies. (**A**) Antisense oligonucleotides (AOs) can degrade target mRNA transcripts by recruiting RNase H, or by inducing steric blocking. AO-mediated steric blocking of a target mRNA transcript can inhibit proper translation, splicing, and/or stability. Further downstream degradation pathways can be initiated on steric-blocked mRNA transcripts. (**B**) RNAi-based therapies (siRNAs, miRNAs, etc.) degrade target mRNA transcripts using the RNA interference (RNAi) pathway. RNAi can be induced by miRNAs or siRNAs complementary to an mRNA transcript. DICER processes the precursor molecules to produce miRNA or siRNAs, which then are loaded into the Argonaute (AGO) protein of the RNA-induced silencing complex (RISC). Using the guide strand, the RISC targets a complementary mRNA transcript and induces translational inhibition, sequestration, and/or mRNA degradation (miRNA-RISC) or simply mRNA degradation (siRNA-RISC).

**Figure 4 ijms-25-09065-f004:**
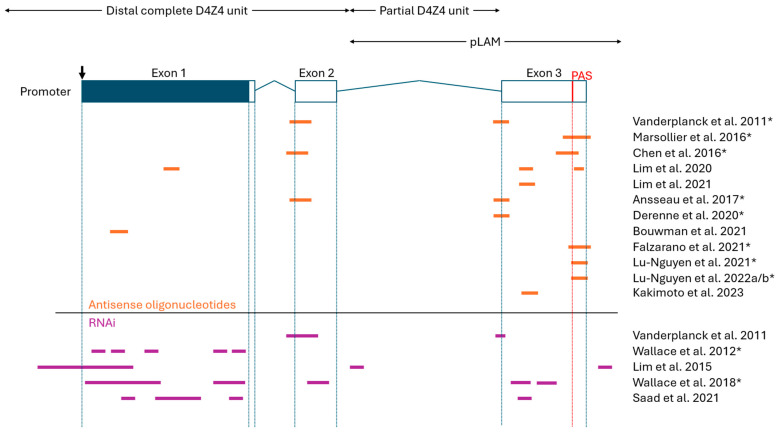
An overview of the oligonucleotide target sites on *DUX4* mRNA. Orange (antisense oligonucleotides) and purple (RNAi) lines indicate oligo target sites. Partially overlapping target sites are simply represented as a continuous line. All attempted target sites are included, even oligos that showed poor indications in corresponding studies. The figure is not to scale and approximate oligo sites/sizes are shown. (Top) A schematic representation of the *DUX4* gene (downwards arrow indicates start codon; blue, open reading frame; boxes, exons; lines, introns; red line, polyadenylation signal). The distal D4Z4 unit, partial D4Z4 unit, and pLAM region are indicated by double-sided arrows. Note that the following groups used some or all of the same oligonucleotides (indicated by *): Vanderplanck et al. (2011), Ansseau et al. (2017), and Derenne et al. (2020) [25,29,30]; Marsollier et al. (2016) and Chen et al. (2016) [27,28]; Marsollier et al. (2016) and Falzarano et al. (2021) [27,34]; Lu-Nguyen et al. (2021), Lu-Nguyen et al. (2022a), and Lu-Nguyen et al. (2022b) [35,36,37]; Wallace et al. (2012) and Wallace et al. (2018) [39,40].

**Table 1 ijms-25-09065-t001:** Overview of preclinical studies investigating antisense oligonucleotides to treat FSHD.

Study	Type	Strategy	Chemical Modification(s)	*DUX4* Target Site	Delivery Mechanism	In Vivo Injection Route	In Vitro Model	In Vivo Model	In Vitro Dose	In Vivo Dose (Total #; Dosing Interval)	*DUX4* Knockdown Results	Other Results
Vanderplanck et al., 2011 [25]	AO	Steric blocking	2′-OMe, PS	Ex2 SA, Ex3 SA	None	N/A	Primary FSHD myoblasts (differentiated post-treatment)	N/A	50 nM (Ex2 SA), 150 nM (Ex3 SA)	N/A	30% (Ex2 SA) and 50% (Ex3 SA) ↓*DUX4* mRNA (3 days post-treatment)	↓TP53 protein
Lim et al., 2015 [26]	Gapmer AO	RNase H degradation	2′-MOE gapmer, PS	Promoter	None	N/A	Primary FSHD myoblasts (differentiated post-treatment)	N/A	N/A	N/A	No ↓*DUX4* mRNA observed	None
Marsollier et al., 2016 [27]	PMO AO	Steric blocking	PMO	Ex3 PAS, Ex3 CS	None	N/A	Immortalized FSHD myotubes	N/A	50 nM	N/A	40% (Ex3 PAS) and 52% (Ex3 CS) ↓*DUX4* mRNA (2 days post-treatment)	↓*TRIM43, ZSCAN4, MBD3L2* expression
Chen et al., 2016 [28]	PMO AO	Steric blocking	PMO	Ex2 SA, Ex3, Ex3 PAS	None	N/A	Primary FSHD myotubes	N/A	10 μmol/L	N/A	Not assessed	↓DUX4+ nuclei (Ex3 PAS only); ↓*TRIM43, ZSCAN4, MBD3L2, CCNA1* expression (Ex3 PAS only)
				Ex3 PAS		Local (i.m., via electroporation)	N/A	FSHD muscle xenograft mice	N/A	20 μg (1×; N/A)	~100% ↓*DUX4* mRNA in FSHD muscle xenograft (14 days post-treatment)	↓*ZSCAN4, MBD3L5* expression
Ansseau et al., 2017 [29]	AO	Steric blocking	2′-OMe, PS	Ex2 SA, Ex3 SA	None	N/A	Primary aFSHD and dFSHD myoblasts (differentiated post-treatment)	N/A	50 nM (Ex2 SA), 10 nM (Ex3 SA)	N/A	~90% (Ex2 SA, Ex3 SA) ↓*DUX4* mRNA (3 days post-treatment)	↓DUX4+ nuclei; ↓aFSHD myotubes; ↓*TRIM43* expression
	Vivo-PMO AO		PMO	Ex3 SA	Vivo conjugate	Local (i.m.)	N/A	AVV-DUX4 mice	N/A	10 μg (1×; N/A)	30-fold ↓*DUX4* mRNA (10 days post-injection)	None
Derenne et al., 2020 [30]	Vivo-PMO AO	Steric blocking	PMO	Ex3 SA	Vivo conjugate	Local (i.m., via electroporation)	N/A	DUX4 IMEP mice		250 μg (1×; N/A)	Not assessed	2.5-fold ↓histological lesion compared to non-treated
Lim et al., 2020 [31]	Gapmer AO	RNase H degradation	LNA gapmer, PS	Ex1, Ex3	None	N/A	Immortalized FSHD myotubes	N/A	100 nM	N/A	~100% ↓*DUX4* mRNA (1 day post-treatment)	↓*TRIM43, ZSCAN4*, *MBD3L2* expression; partial transcriptomic restoration; ↑muscle cell fusion/size
				Ex3	None	Local (i.m.)	N/A	*FLExDUX4* mice	N/A	20 μg (3×; every 2 days)	~84% ↓*DUX4* mRNA in TA (1 day post-treatment)	None
Lim et al., 2021 [32]	Gapmer AO	RNase H degradation	2′-MOE gapmer, PS	Ex3	None	N/A	Immortalized FSHD myotubes	N/A	100 nM	N/A	~100% ↓*DUX4* mRNA (1 day post-treatment)	↓*TRIM43, ZSCAN4*, *MBD3L2* expression; partial transcriptomic restoration; ↑muscle cell fusion/size
				Ex3	None	Local (i.m.)	N/A	*FLExDUX4* mice	N/A	20 μg (3×, every 2 days)	~65% ↓*DUX4* mRNA in TA muscle (1 day post-treatment)	None
Bouwman et al., 2021 [33]	Gapmer AO	RNase H degradation	cEt gapmer, 5′-mC	Ex1	Palmitoyl conjugate	Systemic (s.c.)	N/A	*FLExDUX4* mice	N/A	50 mg/kg (13×; biweekly for first 4 weeks, weekly for next 5 weeks)	Mean 37% ↓*DUX4* mRNA in QUA, TRI, GAS, and TA (7 days post-treatment)	Mean 73% ↓DUX4+ nuclei; >60% ↓*Wfdc3, Agtr2, Serpinb6c* expression in QUA, TRI, GAS, and TA; ↓muscle pathology; ↓inflammation and fibrosis pathways
Falzarano et al., 2021 [34]	PMO AO	Steric blocking	PMO	Ex3 CS	Chitosan-shelled NBs	N/A	Immortalized FSHD myotubes	N/A	50nM	N/A	No ↓*DUX4* mRNA observed compared to naked PMO-CS3 control	Poor release of PMO-CS3 from chitosan-shelled NBs
Lu-Nguyen et al., 2021 [35]	Vivo-PMO AO	Steric blocking	PMO	Ex3 PAS, Ex3 CS	Vivo conjugate	N/A	Immortalized FSHD myotubes	N/A	10 μM	N/A	>50% ↓*DUX4* mRNA	53–68% ↓*TRIM36*, 57–81% ↓*ZSCAN4*, 65–85% ↓*PRAMEF2* expression
						Systemic (i.p.)	N/A	*FLExDUX4* mice	N/A	10 mg/kg (4×; weekly)	~50% ↓*DUX4* mRNA (7 days post-treatment)	~50% ↓*Trim36, Wfdc3* expression; 12% ↓muscle atrophy; 52% ↑*in situ* muscle strength; 17% ↓muscle fibrosis; ↑locomotor activity; 22% ↓fatigue level
Lu-Nguyen et al., 2022a [36]	Vivo-PMO AO	Steric blocking	PMO	Ex3 PAS, Ex3 CS	Vivo conjugate	Systemic (i.p.)	N/A	*FLExDUX4* mice	N/A	10 mg/kg (4×; weekly)	~50% ↓*DUX4* mRNA in DIA	↓DUX4+ nuclei; ~50% ↓*Trim36, Wfdc3* mRNA in DIA; ↓muscle fibrosis; ↑muscle regeneration
Lu-Nguyen et al., 2022b [37]	Vivo-PMO AO	Steric blocking	PMO	Ex3 PAS, Ex3 CS	Vivo conjugate	Systemic (i.p.)	N/A	*FLExDUX4* mice	N/A	10 mg/kg (12×; weekly/biweekly)	60% (in DIA) and 40% (in TA) ↓*DUX4* mRNA (~3.5 days post-treatment)	18% ↑body-wide muscle mass; 32% ↑muscle strength; 29% ↓myofiber central nucleation; 37% ↓muscle fibrosis
Kakimoto et al., 2023 [38]	Gapmer AO	RNase H degradation	ALNA[Ms] gapmer, PS	Ex3	None	N/A	C2C12 cells; Primary FSHD myoblasts (differentiated post-treatment)	N/A	10–3000 nM	N/A	Significant ↓*DUX4* mRNA (measured by luciferase reporter)	↓*TRIM43, ZSCAN4, MBD3L2* mRNA; ↑myotube formation, myogenic fusion, myotube area (>300 nM)
						Systemic (s.c.)	N/A	*FLExDUX4* mice	N/A	15 or 30 mg/kg (3×; every 2 weeks)	~50% ↓*DUX4* mRNA in GAS (30 mg/kg only) (~17 days post-treatment)	↓*Wfdc3* mRNA (15, 30 mg/kg); ↓myofiber central nucleation, plasma CK level (15, 30 mg/kg)
						Systemic (s.c.)	N/A	*FLExDUX4* mice	N/A	10 mg/kg (5×; every 2 weeks)	~40% ↓*DUX4* mRNA in TA (~13 days post-treatment)	↑treadmill running speed, ↑muscle force

Abbreviations: 2′-OMe, 2′-O-methyl; PS, phosphorothioated; PMO, phosphorodiamidate morpholino oligomer; LNA, locked nucleic acid; 2′-MOE, 2′-O-methoxyethyl; cEt, constrained ethyl; 5′-mC, 5′-methylcytosines; ALNA[MS], 2′-N-methanesulfonyl-2′-amino-locked nucleic acid; Ex, exon; SA, splice acceptor; PAS, polyadenylation signal; CS, cleavage site; NBs, nanobubbles; i.m., intramuscular injection; s.c., subcutaneous injection; i.p., intraperitoneal injection; QUA, quadriceps; TRI, triceps; GAS, gastrocnemius; TA, tibialis anterior; DIA, diaphragm; aFSHD, atrophic FSHD; dFSHD, disorganized FSHD; IMEP, i.m. injection and electroporation of naked plasmid DNA; mRNA, messenger RNA; N/A, not applicable. Symbols: Upwards arrow, ↑ = increase in (post-treatment); Downwards arrow, ↓ = decrease in (post-treatment).

**Table 2 ijms-25-09065-t002:** Overview of preclinical studies investigating RNAi-based oligonucleotides to treat FSHD.

Study	Type	Strategy	*DUX4* Target Site	Delivery Mechanism	In Vivo Injection Route	In Vitro Model	In Vivo Model	In Vitro Dose	In Vivo Dose (Total #; Dosing Interval)	*DUX4* Knockdown Results	Other Results
Vanderplanck et al., 2011 [25]	siRNA	RNAi	Ex2 SA, Ex3 SA	None	N/A	Primary FSHD myoblasts (differentiated post-treatment)	N/A	10 nM (Ex3 SA)	N/A	80% (Ex3 SA) ↓*DUX4* mRNA (3 days post-treatment)	↓DUX4, Atrogin1, and TP53 protein levels; ↓MuRF1+ nuclei; ↑muscle size
Wallace et al., 2012 [39]	amiRNA	RNAi	Ex1	AAV	N/A	HEK293 cells	N/A	Unknown	N/A	>50% ↓*DUX4* mRNA (measured by luciferase reporter)	↓DUX4 protein (miR-405, 1156)
					Local (i.m.)	N/A	AVV-DUX4 mice	N/A	3 × 10 particles (1×; N/A)	64% ↓*DUX4* mRNA (2’4 weeks post-treatment)	90% ↓DUX4 protein; improved histopathology; no caspase-3+ myofibers; ↑grip strength
Lim et al., 2015 [26]	siRNA	RNAi	Promoter, Ex1, In2, downstream elements	None	N/A	Primary FSHD myoblasts (differentiated post-treatment)	N/A	N/A	N/A	Up to ~50–90% ↓*DUX4* mRNA	↓DUX4+ nuclei; ↓*ZSCAN4* expression; *DUX4* knockdown is DICER/AGO-dependent
Wallace et al., 2018 [40]	amiRNA	RNAi	Ex1, Ex2, Ex3	AAV	N/A	HEK293 cells	N/A	Unknown	N/A	Up to >75% ↓*DUX4* mRNA (measured by luciferase reporter)	Up to >75% ↓DUX4 protein
			Ex1		Local (i.m.)	N/A	AVV-DUX4 mice	N/A	3 × 10 particles (1×; N/A)	Not assessed	miR-1155 was more toxic than miR-405 (measured by histological analysis)
Saad et al., 2021 [41]	miRNA	RNAi	Ex1, Ex2, Ex3	None	N/A	HEK293 cells	N/A	N/A	N/A	36% (miR-675-5p plasmid) and 91% (lncRNA H19 plasmid, miR-675 precursor) ↓*DUX4* mRNA (measured by luciferase reporter)	↓DUX4-target gene expression; ↓cell death; ↓DNA damage
				None	N/A	Immortalized FSHD myotubes	N/A	N/A	N/A	Significant ↓*DUX4* mRNA	↓DUX4 protein; ↓DUX4+ nuclei; ↓caspase-3/7 activity
				AAV	Local (i.m.)	N/A	AVV-DUX4 mice	N/A	5 × 10 particles (1×; N/A)	56% ↓*DUX4* mRNA in TA (14 days post-treatment)	↓DUX4 protein; 88% ↓*Trim36,* 57% ↓*Wfdc3* expression; 81% ↓myofiber central nucleation
Not fully released, see Arrowhead Pharmaceuticals, Inc. 2021 (press release) [42] and Jagannathan et al., 2021 (report) [43]	siRNA	RNAi	Unknown	Targeted RNAi Molecule (TRiM™)	N/A	Primary FSHD myotubes	N/A	10-100 nM	N/A	Unspecified ↓*DUX4* mRNA	↓DUX4-target gene expression
					Systemic (i.v.)	N/A	*FLExDUX4* mice	N/A	3 mg/kg (1×; N/A)	Significant ↓*DUX4* mRNA	↓DUX4 protein; ↓DUX4-target gene expression; ↓muscle fibrosis; ↑weight; ↑rotarod performance
Not fully released, see Mariot et al., 2022 (abstract) [44]	siRNA	RNAi	Unknown	AAV	Unknown	N/A	*FLExDUX4* mice	N/A	Unknown	High ↓*DUX4* mRNA	↓DUX4-target gene expression; ↓myofiber central nucleation; ↓inflammation
Not fully released, see Malecova et al., 2022 (abstract) [45] and Avidity Biosciences, Inc. 2024 (poster) [46]	siRNA	RNAi	Unknown	anti-mTfR1 mAb conjugate	N/A	Immortalized FSHD myotubes	N/A	Unknown	N/A	Not directly assessed	>75% ↓composite DUX4 and DUX4-target gene expression signature
					Systemic (i.v.)	N/A	*FLExDUX4* mice	N/A	8 mg/kg (1×; N/A)	Not directly assessed	>75% ↓composite DUX4 and DUX4-target gene expression signature in TA (3 weeks post-treatment); ↑treadmill running; ↑muscle force; ↑compound muscle action potential
Not fully released, see Dyne Therapeutics, Inc. 2024a (press release) [47] and 2024b (presentation) [48]	siRNA	RNAi	Unknown	anti-mTfR1 mAb conjugate	N/A	FSHD myotubes (unspecified)	N/A	Unknown	N/A	Not directly assessed	↓mean *TRIM36, ZSCAN4,* and *MBD3L2* expression
					Systemic (i.v.)	N/A	hTfR1/iFLExD mice	N/A	Unknown (1×; N/A)	Not directly assessed	Long-term ↓DUX4-target gene expression in QUA, GAS, and TA; ↓hypotrophic myofibers; ↑treadmill running

Abbreviations: Ex, exon; In, intron; SA, splice acceptor; AAV, adeno-associated virus; mAb, monoclonal antibody; i.m., intramuscular injection; i.v., intravenous injection; QUA, quadriceps; GAS, gastrocnemius; TA, tibialis anterior; AGO, argonaute; mRNA, messenger RNA; siRNA, small-interfering RNA; miRNA, microRNA; amiRNA, artificial microRNA; N/A, not applicable. Symbols: Upwards arrow, ↑ = increase in (post-treatment); Downwards arrow, ↓ = decrease in (post-treatment).

**Table 3 ijms-25-09065-t003:** Overview of preclinical studies investigating other oligonucleotides to treat FSHD.

Study	Type	Strategy	Chemical Modifica-tion(s)	DUX4 Target Site	Delivery Mechanism	In Vivo Injection Route	In Vitro Model	In Vivo Model	In Vitro Dose	In Vivo Dose (Total #; Dosing Interval)	*DUX4* Knockdown Results	Other Results
Himeda et al., 2016 [49]	CRISPR/dCas9-KRAB	CRISPR/dCas9-KRAB-mediated transcriptional repression of *DUX4* (by repressing *DUX4* directly)	N/A	Promoter, Ex1	None	N/A	Primary FSHD myocytes	N/A	N/A	N/A	~55% ↓*DUX4* mRNA (~2 days post-treatment)	↓*TRIM43, ZSCAN4*, *MBD3L2* expression; no off-target effect on expression of D4Z4-proximal genes (*FRG1* and *FRG2*)
Himeda et al., 2018 [50]	CRISPR/dCas9-KRAB	CRISPR/dCas9-KRAB-mediated transcriptional repression of *DUX4* (by repressing *DUX4* epigenetic activators)	N/A	N/A (targets *ASH1L, BRD2, KDM4C, SMARCA5*)	None	N/A	Primary FSHD myocytes	N/A	N/A	N/A	~50% ↓*DUX4* mRNA after activator repression (ASH1L, BRD2, KDM4C, SMARCA5, repressed separately) (3 days post-treatment)	↑H3K9me3 at the D4Z4 array
Klingler et al., 2020 [51]	DNA aptamer	Steric inhibition of DUX4 protein	None	DUX4 protein DNA-binding region (HD1, HD2)	None	N/A	N/A	N/A	N/A	N/A	Not assessed	SELEX-determined DNA aptamers have high affinity to recombinant DUX4 protein
Mariot et al., 2020 [52]	dsDNA	Decoy trapping (decoy binding sites to inhibit DUX4 transactivation)	2′-OMe, PS, HEG linker	N/A (dsDNAs contain DUX4 binding domain)	None	N/A	Primary FSHD myotubes	N/A	50 nM	N/A	Not assessed	39-91% ↓T*RIM43, ZSCAN4* expression
					AAV	Local (i.m.)	N/A	pCS2-mkgDUX4 electrotransfected mice	N/A	10 μg (1×; N/A)	Not assessed	34% ↓*Tm7sf4*, 51% ↓*DuxBl* expression
Himeda et al., 2021 [53]	CRISPR/dCas9-TRD (various domains)	CRISPR/dCas9-TRD-mediated transcriptional repression of *DUX4* (by repressing *DUX4* directly)	N/A	Promoter, Ex1	None	N/A	Primary FSHD myocytes	N/A	N/A	N/A	~30–50% ↓*DUX4* mRNA (3 days post-treatment)	↓*TRIM43*, *MBD3L2* expression
					AAV	Local (i.m.)	N/A	*FLExDUX4* mice	N/A	5e5 particles	~30% ↓*DUX4* mRNA in TA (14 days post-treatment)	Modest ↓*Wfdc3* expression
Das and Chadwick 2021 [54]	CRISPR/dCas9-KRAB	CRISPR/dCas9-KRAB-mediated transcriptional repression of *DUX4* (by repressing *DUX4* directly)	N/A	Ex3	None	N/A	Immortalized FSHD myoblasts	N/A	N/A	N/A	Significant ↓*DUX4* mRNA (sgRNA CR-5A)	↓*TRIM43, ZSCAN4* expression (sgRNA CR-5A); partial ↑H3K9me3 at the D4Z4 array (sgRNA CR-5A)
Rashnonejad et al., 2021 [55]	U7-snRNA expression cassette	Inhibition of pre-mRNA production or maturation (by altering the sn ribonucleoprotein complex to target *DUX4* pre-mRNA)	N/A	Start codon, Ex1 (SA, SD, SE), Ex3 PAS	None	N/A	HEK293 (co-transfected with CMV.DUX4-FL); Immortalized FSHD myotubes	N/A	N/A	N/A	60-95% ↓*DUX4* mRNA (HEK293+CMV.DUX4-FL and immortalized FSHD myotubes)	66–87% ↓DUX4 protein (HEK293+CMV.DUX4-FL); ↓*TRIM43, ZSCAN4, MBD3L2, PRAMEF12* expression (FSHD myotubes); ↓caspase-3/7 activity
Not fully released, see Rashnonejad et al., 2022 (abstract) [56]	CRISPR/Cas13	CRISPR/Cas13-mediated cleavage of *DUX4* mRNA	N/A	Unknown	AAV	Local (i.m.)	N/A	FSHD mice (unspecified)	N/A	Unknown	>50% ↓*DUX4* mRNA	Improved histopathological outcomes; immune response to treatment observed
Not fully released, see Saljoughian et al., 2022 (abstract) [57]	CRISPR/Cas13-ADAR	Cas13-ADAR-mediated *DUX4* mRNA editing (C > U nonsense mutation)	N/A	Unknown	None	N/A	Unknown	N/A	N/A	N/A	Not assessed	sgRNA optimization still ongoing
Sasaki-Honda et al., 2022 (preprint) [58]	CRISPR/dCas9-KRAB and dCas9-D3A/D3A3L	CRISPR/dCas9-(KRAB, D3A/D3A3L)-mediated transcriptional repression of DUX4	N/A	Unknown	None	N/A	FSHD dCas9-KRAB(Neo) iPSCs (differentiated into muscle cells post-treatment)	N/A	N/A (1–3× tEP; every 7 days)	N/A	~50% ↓*DUX4* mRNA (2x and 3x tEP) (26 days post-treatment)	>50% ↓*TRIM43, ZSCAN4, MBD3L2* expression (2x and 3x tEP) (26 days post-treatment)

Abbreviations: 2′-OMe, 2′-O-methyl; PS, phosphorothioated; HEG, hexaethylene glycol; Ex, exon; SA, splice acceptor; SD, splice donor; SE, splice enhancer; PAS, polyadenylation signal; AAV, adeno-associated virus; mAb, monoclonal antibody; i.m., intramuscular injection; TA, tibialis anterior; iPSC, induced pluripotent stem cell; tEP, transient electroporation; SELEX, systematic evolution of ligands by exponential enrichment; dsDNA, double-stranded DNA; mRNA, messenger RNA; snRNA, small nuclear RNA; sgRNA, single-guide RNA; CRISPR, clustered regularly interspaced short palindromic repeat; Cas, CRISPR-associated protein; dCas9, dead (endonuclease deficient) Cas9; KRAB, Krüppel-associated box; TRD, transcriptional repression domain; ADAR, adenosine deaminase acting on RNA; N/A, not applicable. Symbols: Upwards arrow, ↑ = increase in (post-treatment); Downwards arrow, ↓ = decrease in (post-treatment).

## Data Availability

Not applicable.

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
