# Peer review of "Oligonucleotide Therapies for Facioscapulohumeral Muscular Dystrophy: Current Preclinical Landscape"

_ijms, 2024, doi:10.3390/ijms25169065_

Round 1

Reviewer 1 Report

Comments and Suggestions for Authors

A brief but informative review on oligonucleotide therapies for FSHD with a clear focus and carefully defined remit, by a senior author with expertise in both oligonucleotides and FSHD. The figures are generally good and support and extend the text. Good use of tables to present a lot of useful information prevents the text from becoming turgid.

Points for consideration

It should be mentioned that DUX4 is very difficult to even detect in adult muscle from FSHD patients (doi: 10.3390/jpm10030073). Thus outcome measures tend to be indirect and normally DUX4 target genes are assessed but authors use different combinations. Other biomarkers may be more effective. Worth citing this recent review for a balanced overview (doi: 10.15252/emmm.202013695).

Figure 2 nicely highlights the different levels of intervention and the review would be improved if the sections highlighted 1 -6 were then discussed in that order in the text, i.e. gene, pre-mRNA, message, protein levels, even though targeting mRNA is the most prevalent.

Need to mention that DUX4 is restricted to old world primates and so all current animal models are not ideal. They generally induce DUX4 in a genetically modified mouse, so lack the regulatory regions and native expression levels and dynamics. Models are somewhat artificial as in general one is reducing levels of DUX4 that one has initiated. Could also highlight pig model now available (e.g. https://myfshd.org/fshd-therapeutics/). Same with cells lines where should indicate between a primary/immortalized patient line and a myoblast line genetically modified to express DUX4.

Would mention early about deliver methods and how some oligos can be delivered directly but others need mediated delivery such as with AAVs. Maybe in a dedicated section before 2.1?

Tables: do not think that ‘N/A, not applicable’ is a useful term. Items like dose are applicable but were just ‘not reported’.

Line 12 and 36: would consider the non-muscle symptoms ‘rare’.

Line 31: make clear that it is ‘left-right’ asymmetry.

Line 42: would say >100 here and on Figure 1 but ‘1000’ as stated in Figure 1 legend is very high?

Figure 1: would suggest using ‘unaffected’ or ‘healthy’ rather than ‘normal’. Make line for methylation/hypo a little more prominent. Change to ‘Mutation in genes that control epigenetics’. In B would adopt D4Z4 unit as per A with exon 1 and 2 included, as looks like the third one only has exon 1 and 2?

Line 48: add ‘doi: 10.1212/WNL.0000000000007456’.

Line 67: DUX4 is not only expressed in adult muscle (doi: 10.1093/hmg/ddt409) and so there might also be a developmental component, hence might have to intervene with oligos pre-symptoms. DNA methylation is not the only epigenetic change (doi: 10.1146/annurev-genom-083118-014933).

Line 79: distal-most ‘complete’ D4Z4 unit.

Line 91: oligos might not only have to target muscle fibers but also have to access myoblasts in regenerating muscle, so should mention regeneration (doi: 10.1093/hmg/ddaa164).

Line 91: From ‘In this review…’ on would fit better in paragraph below ‘2. Oligonucleotide therapies targeting DUX4’.

Figure 2: As commented, would use this order for the text as easier to follow.

Figure 3: What does box with ‘Etc.’ represent?

Figure 4: Would indicate the partial D4Z4 unit at the top for completeness.

Line 226: Would mention early about deliver methods and how some oligos can be delivered directly but others need mediated delivery such as with AAVs.

Line 233: ‘2.3 Other’ should be expanded into a proper title.

Line 277: change ‘or’ to ‘of’?

Line 296: DUX4 is not only expressed in adult muscle (doi: 10.1093/hmg/ddt409) and so there might also be a developmental component to pathology, hence might have to intervene with oligos pre-symptoms.

Line 299: Skin and bone potentially express DUX4 (see doi: 10.15252/emmm.202013695) and other sites may be involved if targeting DUX4 during development/growth.

Conclusions: should mention new in silico models that are being developed to test therapies directed against DUX4 (e.g. DOI: 10.7554/eLife.88345).

Author Response

Thank you very much for taking the time to review this manuscript. Please find our responses to your comments below. The revised manuscript has been re-submitted with the ‘track changes’ feature enabled should you want to review the changes we made according to your helpful suggestions.

REVIEWER #1

Comment 1: It should be mentioned that DUX4 is very difficult to even detect in adult muscle from FSHD patients (doi: 10.3390/jpm10030073). Thus outcome measures tend to be indirect and normally DUX4 target genes are assessed but authors use different combinations. Other biomarkers may be more effective. Worth citing this recent review for a balanced overview (doi: 10.15252/emmm.202013695).

Response 1: We agree with this suggestion. Thank you. We have made mention of DUX4 being difficult to detect in muscle tissue and the indirect means of evaluating treatment outcome that is used in studies as a result. See section 1.3. However, mention of other biomarkers seems to be outside of the scope of this review, especially considering the reviewed papers generally did not use the PAX7 target gene signature biomarker.

Comment 2: Figure 2 nicely highlights the different levels of intervention and the review would be improved if the sections highlighted 1 -6 were then discussed in that order in the text, i.e. gene, pre-mRNA, message, protein levels, even though targeting mRNA is the most prevalent.

Response 2: While we agree with the sentiment that reordering the mention of each oligonucleotide type according to Figure 2 would be ideal, it would also mean that the table structure needs to be changed entirely. It is our opinion that it is better to keep 3 tables of roughly equal size and preserve the current order in the text. Also, we believe that it is reasonable to mention AO and RNAi approaches first, given that they are the most prevalent and notable out of all targeted oligonucleotide approaches for FSHD. We do not feel that this negatively impacts the readability of the manuscript, especially not to enough of an extent to warrant reordering a large portion of text. Your point, however, is still well-reasoned and we appreciate it nonetheless.

Comment 3: Need to mention that DUX4 is restricted to old world primates and so all current animal models are not ideal. They generally induce DUX4 in a genetically modified mouse, so lack the regulatory regions and native expression levels and dynamics. Models are somewhat artificial as in general one is reducing levels of DUX4 that one has initiated. Could also highlight pig model now available (e.g. https://myfshd.org/fshd-therapeutics/). Same with cells lines where should indicate between a primary/immortalized patient line and a myoblast line genetically modified to express DUX4.

Response 3: We agree with this suggestion. Accordingly, we’ve outlined that DUX4 is restricted to Old World primates, making research models limited in their ability to recapitulate FSHD as it occurs in humans. Thank you for pointing this out. See section 1.3.

Would mention early about deliver methods and how some oligos can be delivered directly but others need mediated delivery such as with AAVs. Maybe in a dedicated section before 2.1?

Response 4: We have made mention of relevant delivery methods as per your suggestions. Please see section 1.4.

Tables: do not think that ‘N/A, not applicable’ is a useful term. Items like dose are applicable but were just ‘not reported’.

Response 5: We replaced ‘N/A’ with ‘None’ where needed. N/A was kept when in parts of the tables where appropriate.

Line 12 and 36: would consider the non-muscle symptoms ‘rare’.

Response 6: While some extramuscular symptoms are rare (cardiac conduction defects), others can be rather common among FSHD patients. For example, retinal vasculopathies and hearing loss has been observed in >50% of patients according to certain case studies (https://doi.org/10.1093/brain/110.3.631 and https://doi.org/10.1002/mus.880181314). Therefore, we believe that stating all non-muscle symptoms as ‘rare’ would be a bit too strongly put.

Line 31: make clear that it is ‘left-right’ asymmetry.

Response 7: Accepted.

Line 42: would say >100 here and on Figure 1 but ‘1000’ as stated in Figure 1 legend is very high?

Response 8: ‘1000’ was a typo. Thank you for catching that. It has been changed to 100+.

Figure 1: would suggest using ‘unaffected’ or ‘healthy’ rather than ‘normal’. Make line for methylation/hypo a little more prominent. Change to ‘Mutation in genes that control epigenetics’. In B would adopt D4Z4 unit as per A with exon 1 and 2 included, as looks like the third one only has exon 1 and 2?

Response 9: Accepted.

Line 48: add ‘doi: 10.1212/WNL.0000000000007456’.

Response 10: Accepted.

Line 67: DUX4 is not only expressed in adult muscle (doi: 10.1093/hmg/ddt409) and so there might also be a developmental component, hence might have to intervene with oligos pre-symptoms. DNA methylation is not the only epigenetic change (doi: 10.1146/annurev-genom-083118-014933).

Response 11: Thank you for bringing these papers to our attention. We agree that these changes are important to avoid including any slightly misleading details regarding the genetics and epigenetics of FSHD. We have removed specific mention of aberrant DUX4 expression in ‘adult’ skeletal muscle given the observed expression of DUX4-fl during fetal development. Similarly, we have changed ‘repressive methylation’ to ‘chromatin repression’ to reflect the more nuanced epigenetic changes outlined in Himeda and Jones’ 2019 paper.

Line 79: distal-most ‘complete’ D4Z4 unit.

Response 12: Accepted.

Line 91: oligos might not only have to target muscle fibers but also have to access myoblasts in regenerating muscle, so should mention regeneration (doi: 10.1093/hmg/ddaa164).

Response 13: We agree, however, we also believe that ‘muscle tissue’ can refer to the regenerating muscle, too. Myogenesis is mentioned elsewhere in the introduction.

Line 91: From ‘In this review…’ on would fit better in paragraph below ‘2. Oligonucleotide therapies targeting DUX4’.

Response 14: Accepted.

Figure 2: As commented, would use this order for the text as easier to follow.

Response 15: See ‘Response 2’.

Figure 3: What does box with ‘Etc.’ represent?

Response 16: ‘Etc.’ was included beside the diagrams of inhibited splicing and translation to suggest that steric blocking can cause other downstream effects as well. Thank you for pointing out the vague wording. This has been replaced with ‘Downstream effects’.

Figure 4: Would indicate the partial D4Z4 unit at the top for completeness.

Response 17: Accepted.

Line 226: Would mention early about deliver methods and how some oligos can be delivered directly but others need mediated delivery such as with AAVs.

Response 18: We agree that the discussion of delivery systems would be a useful addition. Accordingly, we have added a section that briefly covers this. See Section 1.4. Thank you for the suggestion.

Line 233: ‘2.3 Other’ should be expanded into a proper title.

Response 19: Accepted.

Line 277: change ‘or’ to ‘of’?

Response 20: Accepted.

Line 296: DUX4 is not only expressed in adult muscle (doi: 10.1093/hmg/ddt409) and so there might also be a developmental component to pathology, hence might have to intervene with oligos pre-symptoms.

Response 21: Accepted.

Line 299: Skin and bone potentially express DUX4 (see doi: 10.15252/emmm.202013695) and other sites may be involved if targeting DUX4 during development/growth.

Response 22: We agree with this suggestion and have removed this part of the text.

Conclusions: should mention new in silico models that are being developed to test therapies directed against DUX4 (e.g. DOI: 10.7554/eLife.88345).

Response 23: While an interesting new model, we do not believe it warrants mention in the conclusion of this paper.

Reviewer 2 Report

Comments and Suggestions for Authors

The manuscript concerns a review addressing progress in oligonucleotide therapy development for FSHD. Other reviewers have preceded the current submission, whoever, a main strength of this review is the systematic and elaborate overview of published studies provided in table 1. Some points need to be explained in order to improve the text. On line 185, please explain the focus on exons 2 and 3 as target sides. The ongoing clinical trails both concern RNAi strategies. Authors claim AOs are preferred candidates, thus this needs an explanation. In view of the commercial interest of the last author, objective discussion of performance of the different strategies should be more elaborate and founded by citing more published results. Also, in view of the clinical variability of FSHD, the inclusion criteria for patients need to be discussed. The potential synergistic effects of combined strategies is alluded upon but not developed. Add discussion and references in other diseases to back this up.

Comments on the Quality of English Language

English language is fine.

Author Response

Thank you very much for taking the time to review this manuscript. Please find our responses to your comments below. The revised manuscript has been re-submitted with the ‘track changes’ feature enabled should you want to review the changes we made according to your helpful suggestions.

REVIEWER #2

Comment 1: On line 185, please explain the focus on exons 2 and 3 as target sides.

Response 1: As per your comment, we have added an explanation as to why exons 2 and 3 are preferred target sites on DUX4 mRNA. Thank you.

Comment 2: The ongoing clinical trails both concern RNAi strategies. Authors claim AOs are preferred candidates, thus this needs an explanation. In view of the commercial interest of the last author, objective discussion of performance of the different strategies should be more elaborate and founded by citing more published results.

Response 2: We do not recall an instance where we claimed that AOs are preferred candidates over RNAi oligos. Rather, we made mention on line 288 of the original manuscript that all antisense therapies, both AOs and RNAi oligos, are the furthest along in preclinical development. We had also stated that there are more published studies evaluating DUX4-targeting AOs than RNAi oligos (siRNAs, microRNAs). This, however, does not mean to suggest that AOs are preferred, but was simply a comment amount of published work between each oligo type. We have changed our wording to reflect a more neutral tone.  Thank you for pointing this out for us.

Comment 3: Also, in view of the clinical variability of FSHD, the inclusion criteria for patients need to be discussed.

Response 3: We added a sentence that clarifies what is needed for a positive diagnosis for FSHD1 and FSHD2: “FSHD1 is diagnosed by assessing the size of the repeat contraction, while FSHD2 diagnosis may also require testing for mutation in SMCHD1, DNMT3B, and/or LRIF1.”

Comment 4: The potential synergistic effects of combined strategies is alluded upon but not developed. Add discussion and references in other diseases to back this up.

Response 4: We agree that this idea needed to be developed. The synergistic potential of combined treatment strategies for FSHD has been further developed using Duchenne muscular dystrophy as an example. Please see the conclusion section for this change.

Reviewer 3 Report

Comments and Suggestions for Authors

Oligonucleotide Therapies for Facioscapulohumeral Muscular Dystrophy: Current Preclinical Landscape
Samuel L. Beck and Toshifumi Yokota

This is a well written review article for therapeutic approaches with oligonucleotides for Facioscapulohumeral Muscular Dystrophy. It refers many aspects of therapies for this disease, and it is useful not only for basic scientists but also physicians.

I have some suggestions for this review article described below.

1) It would be wonderful if the authors could refer and discuss delivery systems for oligonucleotides briefly.

2) It would be better if the authors could compare with other diseases which have oligonucleotides as a therapeutic approach, such as SMA and DMD.

Author Response

Thank you very much for taking the time to review this manuscript. Please find our responses to your comments below. The revised manuscript has been re-submitted with the ‘track changes’ feature enabled should you want to review the changes we made according to your helpful suggestions.

REVIEWER #3

Comment 1: It would be wonderful if the authors could refer and discuss delivery systems for oligonucleotides briefly.

Response 1: We agree that the discussion of delivery systems would be a useful addition. Accordingly, we have added a section that briefly covers this. See Section 1.4. Thank you for the suggestion.

Comment 2: It would be better if the authors could compare with other diseases which have oligonucleotides as a therapeutic approach, such as SMA and DMD.

Response 2: We have made mention of related diseases where oligonucleotides are also of therapeutic interest (DMD). It can be found in the ‘Conclusions’ section.